# Cold-Pressed Oilseed Cakes as Alternative and Sustainable Feed Ingredients: A Review

**DOI:** 10.3390/foods12030432

**Published:** 2023-01-17

**Authors:** Slađana Rakita, Bojana Kokić, Michele Manoni, Sharon Mazzoleni, Peng Lin, Alice Luciano, Matteo Ottoboni, Federica Cheli, Luciano Pinotti

**Affiliations:** 1Institute of Food Technology, University of Novi Sad, Bulevar cara Lazara 1, 21000 Novi Sad, Serbia; 2Department of Veterinary Medicine and Animal Sciences (DIVAS), University of Milan, 26900 Lodi, Italy; 3CRC I-WE (Coordinating Research Centre: Innovation for Well-Being and Environment), University of Milan, 20133 Milan, Italy

**Keywords:** alternative feed ingredient, ruminants, nonruminants, growth performance, fatty acids, amino acids, nutrient digestibility

## Abstract

Due to the increasing demand for alternative protein feed ingredients, the utilization of oilseed by-products in animal nutrition has been sought as a promising solution to ensure cheap and environmentally sustainable feedstuffs. This review aimed to summarize the nutritional value of six cold-pressed cakes (rapeseed, hempseed, linseed, sunflower seed, camelina seed, and pumpkin seed) and the effects of their inclusion in diet for ruminant, pig, and poultry on nutrient digestibility, growth and productive performance, and quality of the products. The presented results indicated that these unconventional feed ingredients are a good protein and lipid source and have a balanced amino acid and fatty acid profile. However, contradictory results of animal production performances can be found in the literature depending on the cake type and chemical composition, dietary inclusion level, animal category, and trial duration. Due to the substantial amount of essential fatty acid, these cakes can be efficiently used in the production of animal products rich in n-3 and n-6 polyunsaturated fatty acids. However, the utilization of cakes in pig and poultry nutrition is limited because of the presence of antinutritive factors that can deteriorate feed intake and nutrient utilization.

## 1. Introduction

Recently, due to many factors (e.g., excessive use of global resources, increasing global population, unsustainable consumption behaviors, increased demand for animal-source food, and climate change), there is an increased interest in finding alternative protein feeds that can be locally produced to substitute soybean meal in livestock production [1,2,3,4]. For this reason, it is important to focus on circular economy and develop new strategies to make the best use of our resources and eliminate the concept of waste along the food chain. Considering the “waste” produced at different stages of the supply chain, a large amount of biodegradable waste and by-products is produced and discarded every year from the food industry [3,5]. In Europe, the total food waste at the processing stage was found to be 30.6 million tons annually, generated by the various sub-sectors of the processing stage (Figure 1). One solution to reduce food waste is to try to extract the maximum value from food waste and by-products. In general, by-products contain high amounts of nutrients and bioactive compounds, such as carbohydrates, lipids, organic acids, proteins, vitamins, minerals, and antioxidants [6], with numerous health benefits (antitumoral, antimicrobial, and antimutagenic effects) that can be isolated and employed for human food, pharmaceutics, cosmetics, and textiles. Other alternatives after the extraction could be conversion in energy, their use as feedstuff ingredients for livestock production, and production of fertilizers and compost. This review is aimed at providing an overview on the (i) current knowledge about the use of cold-pressed oilseed cakes in animal nutrition; (ii) nutritional and chemical features of six selected cold-pressed oilseed cakes; and (iii) inclusion of cold-pressed oilseed cakes in ruminant, pig, and poultry diet and the effects on growth and productive performance and quality of the products.

## 2. Oilseed By-Products: General Features

Soybean, rapeseed/canola, and sunflower seeds are the most produced in the world, with a production of 337.9, 69.2, and 56.7 million tons, respectively [7,8]. Linseed, camelina, coconut, cotton, pumpkin, and hempseed are also considered as interesting oilseeds in the market [7]. Oilseeds are mostly used as a source of different vegetable oils in animal nutrition. In addition to supplying oil, oil industries started providing by-products rich in protein for animal feeds, because large amounts of by-products are available after the extraction process. Oilseed press cakes and meals are the by-products remaining after the removal of the greater part of the oil from oilseeds. Terms “cake” and “meal” are often used with ambiguity in feed science literature, so these terms should be better defined to avoid misconception. Namely, the term meal refers to the by-product that remains after oil extraction using organic solvents (i.e., hexane, xylene, and toluene), while cake is obtained after using mechanical press method [9]. Mechanical pressing without using heat results in cold-pressed cakes, whereas expeller pressing involves the initial step of conditioning the oilseed using steam prior to pressing and generates expeller-pressed cake [10]. Cold pressing is a cheap, easy, and rapid process to obtain cakes even starting from small quantities of raw materials [9]. Currently, the demand for cold-pressed oilseed cakes obtained with the mechanical press is high because of the simple equipment needed, which can be used directly at the farm site and in rural areas [11]. Instead, solvent extraction methods have some limits, such as cost, toxicity, and the damage to the environment is quite excessive, even though they are used most commonly [9]. Another way to differentiate cakes and meals is their composition, which depends on the oilseed processing type. On one hand, cakes are rich in fiber, residual oils (usually containing more than 10%), and essential fatty acids (FA), but they have a lower protein level than conventionally solvent-extracted meals. On the other hand, meals are rich in proteins but low in residual oil (less than 3%). As a consequence, an innovative local agricultural production chain based on the cultivation of oilseeds was developed. Livestock production systems in the same area require protein and energy supplements from the market. Therefore, the availability of cold-pressed cakes can represent an example of integration between the industry and local livestock production [8,12]. In conclusion, it is possible to think that both oilseed cakes and meals represent a good alternative protein source to cover the increasing demand of protein foods. Indeed, the global demand for animal proteins is expected to double by 2050 [13]. Figure 2 shows the percentage of the main protein sources used in the animal feed sector in Europe, reporting that oilseeds are the main protein sources for animal nutrition in the EU (64%).

## 3. Literature Review Methodology

A systematic literature search was conducted in May 2022 using Google Scholar, and it was limited to scientific articles published in English. The literature search was achieved using the following keywords: (cold-pressed rapeseed cake OR cold-pressed hempseed cake OR cold-pressed linseed cake OR cold-pressed sunflower seed cake OR cold-pressed camelina seed cake OR cold-pressed pumpkin seed cake) AND (chemical composition OR nutritive OR amino acid OR fatty acid OR ruminant OR pig OR hen OR broiler). Articles which were published as full-length articles were selected. Further studies were inspected through searches of reference lists of collected papers and previously published studies. The initial search yielded 91 articles in total. The collected articles were thoroughly evaluated for their potential relevance by carefully reading title, introduction, methodology, and discussion. By analyzing the article content, we concluded that the term “cold-pressed cake” is often misused in the scientific community. The term “cold-pressed cake” is found to be used also for the oilseed by-products which are obtained by screw pressing preceded by conditioning to enhance oil separation (expeller pressed cakes). Furthermore, cold-pressed cakes are also misnamed as oilseed meals but, according to provided methodology in the literature and information about oil extraction conditions and chemical composition of meals, particularly oil content, it was possible to conclude that authors used cold-pressed cakes instead of meals in their research, which classified mentioned studies to be included in the present review. On the other hand, we excluded those articles that used expeller-pressed cakes in their studies (*n* = 7) and cold-pressed cakes that were subsequently subjected to the thermal process of pelleting before animal feeding (*n* = 3). Articles which did not explicitly state that the cakes were obtained by cold-pressing method were also excluded from the scoping review (*n* = 29). After applying exclusion criteria and eliminating the articles, in total, 52 full-length articles related to the application of selected cold-pressed cakes in animal nutrition were included in the review portfolio and considered for further analysis. The collected literature was systematically organized and classified according to the type of cold-pressed cake and animal species. According to our literature analysis, cold-pressed rapeseed cake was the most commonly investigated oilseed cake as an animal feed ingredient, while cold-pressed pumpkin seed cake was the least studied cake.

## 4. Nutritional Composition of Cold-Pressed Oilseed Cakes

Research studies included in the present review are those which used cold-pressed oilseed cakes, where this was expressly stated or could be concluded based on description of obtainment method. Therefore, this review focuses on six cold-pressed oilseed cakes: rapeseed (RSC), hempseed (HSC), linseed (LSC), sunflower (SFC), camelina (CSC), and pumpkin seed cake (PSC). The chemical composition of selected cold-pressed oilseed cakes is presented in Table 1.

Cold-pressed oilseed cakes showed to be a great source of protein. Cakes have high protein content ranging from 19.4% to 62.3%, which is expected, since cold pressing is aimed at retaining the protein in the cake while removing the oil from oilseeds. The most variable and the highest average protein content (50.3%) is in PSC in comparison to other cakes. Cold-pressed cakes are characterized by high oil content ranging from 8.9% to 36.2%. These results clearly demonstrate that a considerable amount of oil remains in cake after mechanical pressing of oilseeds. The cold-pressed cakes retain a higher amount of oil in comparison to solvent-extracted meals. Because of high oil content, cold-pressed cakes contribute a considerable amount of energy to the animal diets. Observed cakes have shown low ash content, which may vary between 4.2% and 8.1%. Fiber content varies from 6.5% for RSC to 37.0% for SFC. A considerable amount of fiber is observed in HSC and PSC. Care should be taken when including feed ingredients rich in fiber, because high fiber content in the diet for monogastric animals may affect animal performance, digestion, and liver lipid metabolism [27]. NDF and ADF content are in the range of 17.0–53.4% and 5.1–39.2%, respectively, whereas PSC has lower average NDF as well as ADF content compared to other cakes. The values of fiber, NDF, and ADF content are in line with the work of Serrapica et al. [40], where various oilseed cakes were shown to be valuable alternative protein sources for ruminant nutrition. RSC and CSC are reported to contain significant concentration of antinutritive compounds glucosinolates, which may negatively affect palatability and decrease animal performance, especially in monogastric animals [37]. The concentration of glucosinolates is notably higher in CSC than in RSC, and it varies between 8.8 and 16.9 µmol/g for RSC and 22.9 and 46.1 µmol/g for CSC. However, it is believed that glucosinolates from CSC have less deleterious effects than glucosinolates from RSC, as its metabolites do not contain progoitrin, which can be transformed into toxic goitrin, which is the case with RSC. On the other hand, glucosinolates from CSC have been proposed to have beneficial effects on health due to their chemo-protective and anticarcinogenic activity [41].

Amino acid (AA) profile, particularly essential AA (EAA), determines the biological value of proteins. The content of EAA in the cakes was in the range of 8.9–15.6%, as presented in Table 2. The observed cakes are rich in EAA, which makes these feed ingredients a valuable source of protein and AA in ruminants and nonruminants. Arginine and leucine are the major EAA observed in all cakes, of which HSC has the highest level of arginine (4.1%), followed by CSC and LSC (3.0%). CSC and HSC have the highest level of leucine (2.4 and 2.3%, respectively). Arginine is an important AA for livestock production, as dietary supplementation with rumen-protected arginine can support both growth and production performance in ruminants [42]. Methionine, lysine, and leucine are often limiting EAA in animal nutrition. Methionine is a deficient AA for all cakes, ranging from 0.3 to 0.88%. The content of lysine is in the range of 1.4–2.1%. Observed cakes are also rich in nonessential AA (NAA) (12.6–22.3%), of which glutamic acid was the dominant acid (3.7–7.8%), followed by aspartic acid (1.7–3.5%). Some studies reported a beneficial effect of glutamic acid on growth performance of weaning and growing pigs [43,44]. There are no available data on the AA profile of SFC and PSC in the reviewed literature and, therefore, AA composition of these cakes is not debated. AA profile of HSC is more similar to that in hempseed meal than in whole hempseed, dehulled hempseed, or hemp hulls [45]. Furthermore, RSC has AA composition comparable to that in rapeseed cake, full-fat rapeseed, and canola meal [46].

Fatty acid (FA) profile of cold-pressed cakes is shown in Table 3. The results show that the cakes are characterized by a low amount of saturated fatty acids (SFA), of which palmitic acid is the most dominant. PSC has the highest average share of palmitic and stearic acid (12.8 and 5.1%, respectively). The highest average content of oleic acid has been reported for RSC (55.6%), while the lowest level was found in SFC (10.3%). HSC, LSC, CSC, and PSC are a major source of polyunsaturated fatty acids (PUFA), particularly linoleic acid (LA) and α-linolenic acid (ALA), which are considered essential. These FA are vital for health and must be included in the diet because the body cannot synthesize them. The major sources of LA are HSC and PSC (54.3 and 50.1%, respectively). LSC has the highest share of ALA (51.5%), while CSC is observed as the second highest source of ALA among all oilseed cakes (22.0–35.0%). The lowest ratio of n-6 and n-3 fatty acids have LSC and CSC, whereas SFC and PSC are characterized by significantly higher n-6/n-3 ratio.

## 5. Cold-Pressed Oilseed Cakes in Ruminant Nutrition

### 5.1. Effects on In Vitro, In Vivo, and In Situ Parameters

By evaluating the dietary inclusion of cold-pressed cakes in beef cattle diet on in vitro rumen fermentation, Benhissi et al. [54] demonstrated the potential of SFC to alter microbial rumen fermentation (reduced total volatile fatty acids (VFA) and methane production) at a total fat level of 60 g/kg DM in comparison to palm fat. However, the same was not obtained for RSC. The main reason for this can be attributed to the different FA profile of tested fat sources. The predominant FA in palm fat and RSC are SFA and monounsaturated fatty acids (MUFA), respectively, whereas SFC is a rich source of PUFA (mainly LA). In another in vitro study, Benhissi et al. [55] examined the effects of replacing palm fat with RSC or SFC on FA biohydrogenation using an artificial rumen. While RSC had no effect on nutrient disappearance and rumen fermentation, SFC reduced dry matter (DM), organic matter (OM), crude protein (CP), NDF, and ADF disappearances. Moreover, SFC decreased total VFA and methane production and shifted rumen fermentation pattern towards lower acetate and higher propionate proportion. Both RSC and SFC decreased total SFA in rumen effluent and increased the content of vaccenic acid (VA), with the magnitude of change being higher for SFC, while the increase in cis-9, trans-11 conjugated linolenic acid (CLA) was noted only for SFC. McKinnon and Walker [56] determined the in situ degradation characteristic of RSC from biodiesel production and canola meal obtained using solvent and heat. The extent of rumen degradation of CP (773 g/kg), ADF (428 g/kg), and NDF (503 g/kg) was greater for RSC in comparison to canola meal, indicating that a larger portion of RSC is degraded in the rumen. Authors suggested that the use of heat in canola meal production may serve to bind some of the protein and fiber, making it less available for the degradation in the rumen. In an in situ experiment using HSC, Karlsson and Martinsson [57] reported a value of 709 g/kg CP for effective protein degradation in lactating cow rumen. However, intestinal digestibility of protein (307 g/kg RUP) was remarkably low, which consequently led to a high amount of indigestible CP in HSC (202 g/kg CP), making it a poorer protein supplement in comparison to peas or heat-treated RSC. Moloney et al. [22] determined in vivo OM and CP digestibility of CSC (722 and 771 g/kg, respectively) and, based on the obtained results, concluded that CSC was a superior protein source to sunflower meal or copra meal and comparable with corn gluten. In an in situ experiment conducted by Lawrence and Anderson [58], CSC was evaluated as a protein feed ingredient in dairy cattle nutrition and compared with other common protein sources. CSC exhibited the highest content of rumen degradable protein (764 g/kg of CP) and a total digestible protein (955 g/kg of CP) comparable to linseed meal and soybean meal. All the above in vitro, in vivo, and in situ results represent initial research that demonstrates the potential of cold-pressed oilseed cakes to be included in ruminant nutrition as alternative protein sources of rumen degradable nutrients. Based on the previously published results of Benhissi et al. [54] and Benhissi et al. [55] regarding the reduction in microbial rumen fermentation upon inclusion of high-linoleic cakes, follow-up of this initial research is feeding trials, which are necessary in order to determine whether selected oilseed cakes high in PUFA might lead to impaired rumen function.

### 5.2. Effects on Animal Production Performance

Appendix A summarizes available studies and the main effects when cold-pressed oilseed cakes are included in ruminant nutrition. Studies evaluating the influence of cold-pressed oilseed cakes’ inclusion in cattle diet on growth performance are rather scarce. To the authors’ best knowledge there are no available studies that examined the effects of SFC, LSC, and PSC inclusion in growing cattle diet. Few available studies reported that inclusion of RSC, HSC, and CSC in steers, heifers, and lamb diet did not exhibit detrimental effects on animal growth performance [1,36,38,52,57]. No changes in ADG or DMI were recorded for RSC [36] and HSC [57] supplemented diet in comparison to barley-based diet without protein supplement, HSC in comparison to soybean meal [1], as well as CSC in comparison to DDGS [52] and DDGS or linseed meal [38]. An increase in DMI was noted only for calves fed HSC in comparison to a soybean meal, while the effect was not observed in steers [1]. Although no statistical differences were obtained for ADG and DMI but, rather, numerical, feed efficiency tended to be lower in calves [1,36] and heifers [38] but not in lambs [57]. Based on only a few available studies—reported above—it can be noticed that the inclusion of cold-pressed oilseed cakes does not have any detrimental effect on cattle growth performance, which opens new opportunities for their use.

The possibility of cold-pressed oilseed cake utilization in ruminant nutrition has been more investigated in studies with dairy ruminants (cows, sheeps, and goats). Due to a large number of available studies, detailed data regarding the inclusion level of examined cold-pressed oilseed cakes, as well as the composition of the control diet, are presented in Appendix A, while only general effects will be discussed in the text below. An increase in milk yield was recorded in studies where cows’ diet was supplemented with RSC [48,59,60], HSC [2], and LSC [59], while other studies found no statistically significant changes in milk yield for supplementation with RSC [12,61,62], LSC [63], SFC [64], or CSC [65]. On the other hand, Rinne et al. [20] reported a linear decrease in milk production when heat-moisture-treated RSC was gradually (32, 64, and 92 g/kg DM) replaced with cold-pressed LSC in a grass silage-based diet for cows. Authors proposed several potential causes for the reduced production potential of LSC in comparison to heat-moisture-treated RSC, including the slightly higher lipid content of LSC, which may have resulted in decreased diet digestibility, differences in FA composition, more extensive ruminal degradability of LSC, differences in the AA profile, and, possibly, antinutritional factors contained in LSC. However, in another study, LSC caused an increase in milk yield ranging around 13% in comparison to soybean meal as a nonfat control diet [59]. In the same study, a decrease in milk yield was recorded only for diet supplemented with CSC (approximately 12%). In contrast to cows, sheep milk yield was mostly unaffected when RSC [66,67,68] and SFC [66,68] were included in animal diet. The same was observed in dairy goats for HSC [25] and PSC [15] inclusion. Only two studies on small ruminants reported significant improvement in milk yield ranging around 5% for HSC-supplemented dairy sheep diet [49] and around 27% for LSC-supplemented dairy goat diet [69]. When it comes to inclusion of cold-pressed oilseed cakes and their influence on milk protein content, results are generally uniform. A decrease and tendency to decrease milk protein content was recorded in lipid-supplemented dairy cow diets [2,20,48,59,60,61]. Lerch et al. [61] suggested that this could be a result of a dilution effect, since lipid supplementation often increases milk production without changing protein yield. This assumption seemed to be accurate because no decrease in milk protein content was reported only in studies where no increase in milk yield was obtained [12,62,63,64,65]. As already mentioned, sheep and goat milk yields were mostly unaffected by lipid supplementation, and so was the milk protein content [15,25,49,66,67].

A milk component that has a major influence on raw material processing and is a carrier of taste and aroma is milk fat. Markiewicz-Kęszycka et al. [70] stated that the proportion of fat in cow’s milk is typically 3.3–4.4%, while goat’s and ewe’s milk contains approximately 3.25–4.2% and 7.1% of fat, respectively. Dietary lipid metabolism in ruminants, as well as milk and meat fat biosynthesis are complex processes outside the scope of this review; therefore, readers are referred to other literature for more detailed data [71,72,73,74]. Briefly, when dietary lipids enter the rumen, the initial step is the extensive hydrolysis carried out by rumen bacteria, which results in the release of free FA [75]. Second major transformation of dietary fats in rumen is biohydrogenation of unsaturated FA (UFA) to SFA (particularly to stearic acid). Only few species of rumen bacteria are involved in this process, which is a protective mechanism against the toxic effects of UFA [76]. According to Fredeen [77], lipid supplements rich in PUFA can have an antimicrobial effect, resulting in reduced fiber digestion, reduced acetate:propionate ratio, and depressed milk fat synthesis. Production of milk with a low fat content is often detrimental, both financially and for many manufacturing processes. Implications regarding the inclusion of cold-pressed oilseed cakes in dairy ruminant nutrition on milk fat percentage are rather inconsistent. Some authors reported that there was no change in milk fat percentage when RSC [12,60,61,62], LSC [63], SFC [64], and CSC [65] were included in dairy cows’ diet. Others reported a decrease for RSC [48], HSC [2], LSC [59], and CSC [59] supplemented diets. According to Dewanckele et al. [78], inclusion of lipid supplements in ruminant diets might increase the incidence of diet-induced milk fat depression (MFD) in dairy cattle and, less frequently, in small ruminants. MFD is a consequence of inhibition of *de novo* FA synthesis by some PUFA ruminal biohydrogenation intermediates [76]. In contrast to cows, MFD is not common in goats and sheep. Indeed, inclusion of cold-pressed oilseed cakes in small ruminant nutrition did not exhibit a detrimental effect on milk fat content [15,25,66,67,68,69]. There is only one available study where SFC included in sheep diet caused a significant decrease in milk fat, presumably because of the high amount of LA present in SFC, as Amores et al. [66] observed. In contrast, in a similar study, the inclusion of SFC did not lead to milk fat depression, likely due to a lower PUFA content of the cake used [68].

### 5.3. Effects on FA Profile of Milk and Meat

Milk fat is one of the most complex natural fats that is estimated to have over 400 different FA. In ruminant milk, SFA account for 60–70%, MUFA 20–35%, while PUFA account for as little as 3% of all FA [70]. In ruminants, milk FA arise from two sources—*de novo* synthesis in the mammary gland and the mammary uptake of preformed long-chain FA [79]. Although ruminant diet contains a high proportion of UFA, due to extensive hydrolysis and biohydrogenation in the rumen, the FA that reach the small intestine are mainly free SFA. However, there are some biohydrogenation intermediates that can escape from the rumen and two of the major ones are VA and cis-9, trans-11 CLA. VA is formed from both LA and ALA, while cis-9, trans-11 CLA is formed during the biohydrogenation of only LA. Milk CLA is derived from rumen CLA but the major source is from endogenous synthesis of CLA in the mammary gland from VA [48,80]. Consequently, many studies have been focused on obtaining animal products with greater added value, such as milk and dairy products with increased concentrations of FA that have a beneficial effect on human health. Factors affecting FA profile of milk can be divided into biological (breed, cow individuality, milk yield, lactation stage, and parity) and external (feeding ration composition and management). Although all of the above-mentioned factors affect the FA profile of milk, the main factors are related to nutrition. The inclusion of cold-pressed oilseed cakes in ruminant nutrition can be a proper tool to improve FA profile of milk. All cold-pressed oilseed cakes covered by this review are rich sources of UFA. The inclusion of these cakes in dairy ruminant nutrition can potentially influence the ruminal biohydrogenation process, which will, in turn, lead to changes in FA profile of milk. A more beneficial profile of FA from a nutritional point of view refers to a lower proportion of SFA and a higher proportion of UFA, with the emphasis on n-3 PUFA, and a lower n-6/n-3 ratio [15,81]. Based on the available literature, effects of cold-pressed oilseed cakes’ inclusion in dairy ruminants’ nutrition on FA profile of milk are inconsistent. Reduction in SFA and increase in MUFA and PUFA content in cow’s milk were reported in several studies where RSC [48,59,82], LSC [59], and CSC [59,65] were included in the diet, while others reported no change for RSC [12], LSC [63], and SFC [64] supplemented diet. On the other hand, Rinne et al. [20] reported that gradual replacement of heat-moisture-treated RSC with LSC led to an increase in SFA, with only a minor effect on cow’s milk UFA. Studies with small ruminants presented a much clearer outcome in terms of desired FA changes in milk. Beneficial effects were noted for RSC [66,67,68], HSC [49], and SFC [66,68] supplemented sheep diet and LSC [69] and PSC [15] supplemented goat diet. No changes in SFA and PUFA and minor decrease in MUFA were reported only by Šalavardić et al. [25] for goats fed HSC. Several studies covered by this review reported that the concentration of CLA and VA in cow’s milk fed a control diet was in a range of 0.4–0.8% and 1.0–1.4%, respectively [12,48,59,64,65]. Goiri et al. [12] found no increase in CLA and VA content when RSC was used in the diet. In contrast, increase in CLA ranging 35–75% and VA ranging 31–127% was noted by others when cow diet was supplemented with RSC, LSC, SFC, and CSC [48,59,64,65]. The highest concentrations of CLA (4.9%) and VA (8.3%), which were around 7.5 times higher than those in the control group, were recorded in a study by Mihhejev et al. [59] when cows were fed CSC. However, this notable increase was not repeated with CSC-supplemented diet in a study by Toma et al. [65]. Similar results were also obtained for sheep milk. Two studies reported only minor changes in milk CLA and VA concentration for RSC-supplemented sheep diet [67,68]. Other studies obtained CLA-enriched milk (33–155%) and VA-enriched milk (68–163%) upon inclusion of RSC [66], HSC [49], and SFC [66,68]. Available studies regarding the inclusion of cold-pressed cakes in goat diet and possible increase in CLA and VA are rather scarce. Jozwik et al. [69] reported a 10-fold increase in CLA for LSC-supplemented diet, while Boldea et al. [15] noticed only a minor increase (around 10%) for PSC-supplemented goat diet. Based on the above results, it can be noticed that RSC has the lowest potential of all cold-pressed cakes towards CLA- and VA-enriched milk. This was confirmed by previous studies by Mihhejev et al. [59], Amores et al. [66] and Pascual et al. [68]. In these studies, SFC, LSC, and CSC all exhibited a significantly higher increase in milk CLA and VA in comparison to RSC. This finding could be explained by the fact that RSC is rich in oleic acid, which is not a precursor of either CLA or VA. In contrast, SFC, LSC, and CSC are excellent sources of either LA or ALA that are biohydrogenated in rumen to CLA or VA. According to dietary recommendations, the ratio of n-6/n-3 fatty acids of 1–4:1 has been considered desirable to maintain human health and combat certain chronic diseases. Several studies covered by this review reported that cow’s milk n-6/n-3 ratio for cows fed a control diet was in a range of 3.7–8.2 [12,63,64,65]. It is evident that this range is quite wide and that it is probably an effect of various factors, such as breed, lactation stage, as well as the composition of control diet. No change in milk n-6/n-3 ratio was recorded by Goiri et al. [64] for SFC-supplemented diet. However, a previously published study reported that inclusion of RSC as a replacement for conventional feedstuffs yielded milk with lower n-6/n-3 ratio (4.2 vs. 4.7) [12], and the same was recorded for cows fed LSC compared to RSC (4.3 vs. 8.0) [63]. However, Toma et al. [65] observed that n-6/n-3 ratio in cow’s milk fed a control diet was 8.2 and that a partial or complete substitution of sunflower meal with CSC caused an increase to 9.4 and 9.9, respectively. Šalavardić et al. [25] reported a decrease in goat milk n-6/n-3 ratio from 6.2 to 5.2 when HSC partially replaced soybean meal and extruded soybean in a control diet. In contrast, Boldea et al. [15] noted that the replacement of sunflower meal with PSC led to an increase in milk n-6/n-3 ratio from 8.1 to 10.4, presumably because of the high amount of LA present in PSC that contributed to a significant increase in total n-6 FA in milk. In contrast to cow’s and goat’s milk, n-6/n-3 ratio in sheep’s milk is lower. In a study carried out by Pascual et al. [68], RSC or SFC were included in sheep diet as the only fat source (instead of soybean meal and hydrogenated palm fat). Feeding RSC together with sainfoin hay (contains condensed tannins) reduced milk n-6/n-3 ratio from 1.8 to 1.7, while SFC caused an increase from 1.8 to 4.1. The authors suggested that this result is an effect of various FA profiles of used cold-pressed cakes, namely, RSC is high in oleic acid, while SFC is high in LA. Mierliță [49] also reported a decrease in sheep’s milk n-6/n-3 ratio from 1.9 to 1.5 after inclusion of HSC as a partial substitute for sunflower meal.

Studies examining the effects of cold-pressed oilseed cakes on meat FA profile are limited. He et al. [36] investigated the effects of substituting RSC for barley grain on meat FA profile of feedlot cattle. Inclusion of 30% RSC in both growing and finishing diets resulted in higher content of total PUFA (4.4 vs. 3.4%) and n-3 PUFA (0.7 vs. 0.5%) and a decrease in n-6/n-3 ratio (3.3 vs. 4.5) in the *pars costalis diaphragmatis* muscle as compared with the control diet, while no changes in SFA and MUFA were noted. Content of meat CLA was increased from 0.3 to 0.5%, whereas only a minor increase was recorded for meat VA content (1.1 vs. 0.8%). In a study conducted by Turner et al. [83], the inclusion of HSC in comparison to soybean meal improved FA profile in *M. longissimus dorsi* in terms of increased content of CLA (0.2 vs. 0.1%), VA (0.6 vs. 0.3%), and MUFA (51.4 vs. 48.4%), and a decreased n-6/n-3 ratio (4.1 vs. 5.0), while no changes in the content of SFA, PUFA, n-6, and n-3 FA were noted. To the authors’ best knowledge, there is only one available study with lambs where the effects of feeding HSC on meat FA profile were examined. The experimental diet contained 218 g/kg DM HSC and was compared with a barley-based diet without protein supplement [84]. Authors reported only minor changes on lamb meat, such as a tendency for the HSC diet to increase meat PUFA and n-6 FA. However, changes in meat CLA, VA, SFA, MUFA, n-3, as well as n-6/n-3 ratio were not obtained. The lack of diet effects authors attributed to extensive ALA biohydrogenation when feeding the HSC diet to lambs.

### 5.4. Effects on Other Parameters

A shift towards greater content of UFA in milk fat can lead to off flavors due to FA oxidation. However, several studies reported that there were no negative effects in terms of milk and cheese sensory properties [48,59,67,68]. Mierliță [49] found a significant increase in content of α-tocopherol in sheep milk when HSC was included in the diet and concluded that hemp has a good potential as a natural antioxidant and could contribute to preventing lipid oxidation in raw milk. Another beneficial aspect of using cold-pressed oilseed cakes is a reduction in enteric CH_4_ emissions. Moate et al. [60] concluded that, regardless of type of fat supplementation, for each 10 g/kg DM increase in dietary fat concentration, enteric emissions are reduced by 3.5%. Hessle et al. [1] reported that HSC as a protein feed in growing cattle compared to soybean meal resulted in improved rumen function due to higher fiber content and/or lower starch content. Negative effects on animal health and fertility were not observed by Johansson and Nadeau [48] when conventional protein source was replaced with RSC in dairy cow diet. The same was reported by Jóźwik et al. [63] and, for HSC-supplemented goat diet, by Šalavardić et al. [25]. Inclusion of CSC as a protein source in heifer diet had no adverse effects on animal reproductive performance in comparison to DDGS [52]. In another study conducted by Lawrence et al. [38], feeding CSC to growing dairy heifers resulted in unmodified rumen fermentation characteristics, as well as blood metabolites and metabolic hormones, in comparison to DDGS and linseed meal.

## 6. Cold-Pressed Oilseed Cakes in Pig Nutrition

### 6.1. Effects on Nutrient Digestibility

The main effects of utilization of cold-pressed oilseed cakes in pig diets are presented in Appendix A. The research data suggest that digestible nutrient content of cold-pressed oilseed cakes varies vastly depending on the type of the cake, variety, agronomic conditions, chemical composition, differences in pressing conditions, as well as the employed methodology [18]. Almeida et al. [37] showed that the standardized ileal digestibility (SID) of CP for CSC fed to growing pigs was in the range of 64.1–70.4%, which was similar to that reported by Kahindi et al. [47] (64.8%). The SID of CP for RSC was reported to be in the range from 70.0 to 81.7% [10,18,35]. LSC and HSC showed higher SID of CP (85.4 and 85.2%, respectively) than RSC and CSC [18]. HSC and LSC had the highest average SID of EAA, 87.8 and 85.1%, respectively [18]. According to Woyengo et al. [23], the average SID of EAA for CSC (54.8%) was close to the average value (57.9%) reported by Kahindi et al. [47]. In contrast, higher values of SID of EAA were observed by Almeida et al. [37] for CSC (68.6–77.7%). The average SID of EAA for RSC was in the range of 75.1–84.2% [10,18,35]. The reported values were higher than those observed by Seneviratne et al. [16] (62.5–64.6%). In the same study, rapeseed was pressed using a single-screw press operated with two screw speeds, i.e., slow and fast, and two barrel temperatures (nonheated and heated, where resulting cake temperatures were 53 and 60 °C, respectively). It was demonstrated that the processing conditions had a significant effect on AA quality, which was reflected in changes in the SID of AA. Generally, barrel heating increased the SID of all AA. The SID of Lys for HSC and LSC was 85.3 and 82.0%, respectively, whereas the SID of Met for HSC and LSC was 91.8 and 87.8%, respectively [18]. The SID of Lys and Met for RSC varied between 68.0 and 87.3% and between 81.8 and 86.5%, respectively [10,18,35]. The SID of Lys and Met for CSC was in the range 68.0–72.1% and 75.5–84.0%, respectively, according to Almeida et al. [37]. It was found that RSC had lower SID of Lys in comparison to that in whole rapeseed and rapeseed meal, while the SID of Met in RSC was not affected by pressing conditions [16]. The SID of Lys of RSC increased by 21% when a nonheated barrel with a fast screw speed was used but was not affected in a heated barrel [16]. When barrel speed increased from slow to fast in a nonheated barrel, the SID of Met was decreased by 6.5%, while fast screw speed in a heated barrel increased the SID of Met by 12%. In this study, the increase in the SID of AA was not attributed to the increased CP content but to changes in crude fiber content. Namely, crude fiber content of RSC was lower for a heated barrel than for nonheated, and lower fiber content increased digestibility of proteins and AA [16]. High fiber content can depress the digestibility of AA because it binds AA and, thus, hinders AA absorption [47]. In line with these results, Woyengo et al. [10] demonstrated that the SID of all AA was greater for rapeseed expeller (conditioned and pressed) than that for RSC. This maybe was due to exposure of rapeseed expeller to heat treatment that disrupted call walls and caused denaturation of protein, increased availability for digestion, and, therefore, increased absorption of AA. Additionally, Almeida et al. [37] explained that the differences in the nutrient digestibility of AA among camelina products (seed, expeller, and meal) were not the result of heat treatment but due to the presence of antinutritional factors (phytates and condensed tannins) in the camelina products. A high level of phytate may lower digestibility of AA through adsorption of AA to phytate and inhibit activity of digestive enzymes [47]. Likewise, high content of condensed tannins can depress AA digestibility, since they are known to form indigestible protein complexes with enzymes [47].

Total tract apparent digestibility (TTAD) of gross energy (GE) for HSC, LSC, and RSC was 86.8, 91.5, and 90.2%, respectively [10,18,35]. Notably lower values of TTAD-GE for RSC (64.2%) were observed by Woyengo et al. [10]. It was reported that the TTAD-GE in RSC was 36% higher in a heated barrel of the press in comparison to a nonheated barrel [16]. Apparently, the application of the heat to the barrel during oil extraction led to an increase in TTAD of energy due to the rise in lipid solubility, which might affect improvement in energy digestibility of pressed cake [16]. The TTAD-GE for CSC ranged between 72.4 and 82.0% [23,29,47]. The net energy (NE) and digestible energy (DE) for CSC were noted to be in the range of 9.3–10.7 MJ/kg and 15.3–17.5 MJ/kg, respectively [23,29,47], while NE and DE for RSC were reported to be 8.6–11.9 MJ/kg and 13.2–16.5 MJ/kg, respectively [10,16]. The differences in energy values among oilseed cakes were attributed to differences in the species of oilseed and their chemical composition, as well as the conditions of pressing employed [29]. Cold pressing is less efficient in oil extraction from oilseeds compared with expeller pressing and solvent extraction and, thus, more residual oil is contained in remaining cakes. Therefore, cold pressing yields cakes with greater energy value associated with their remaining oil content which they supply in swine diets [47]. The cold-pressing technology and conditions that are used significantly influence the chemical composition and value of the press cake, which can vary greatly. Namely, increasing screw speed in a nonheated barrel increased the DE and NE value of RSC by 19 and 24%, respectively, whereas fast screw speed in a heated barrel decreased the NE by 10% [16]. More precisely, with the increase in the screw speed in a nonheated barrel, oilseeds were less exposed to mechanical pressing and crushing, which resulted in more residual oil in the cake and higher energy value. In contrast, fast screw speed in a heated barrel facilitated the disruption of cell walls of the seed and releasing of oil that is embedded in cell walls, which resulted in a decrease in residual oil in cakes and lower energy values [16]. Additionally, Kim et al. [29] stated that different BW of pigs in the experiments may also contribute to the different energy values of cold-pressed cakes, as energy digestibility increases with increasing BW, especially in young pigs.

### 6.2. Effects on Growth Performance

Published data on the influence of dietary inclusion of cold-pressed cakes on pig growth performance parameters were observed only for RSC and CSC but not for HSC, LSC, SFC, or PSC. Kaczmarek et al. [39] investigated the influence of partial replacement of soybean meal and soybean oil with RSC with increased oil content, extruded rapeseed cake with increased oil content, or toasted full-fat soybeans in the diet for growing–finishing pigs. After 90 days of feeding, it was found that pigs receiving diet with RSC were characterized by lower ADG and higher feed conversion ratio (FCR) than the remaining animals. The deteriorating production performance of pigs was attributed to the high concentration of glucosinolates in RSC, while RSC subjected to hydrothermal treatment of extrusion lowered the total glucosinolate concentration by approximately 17%. The concentration of total glucosinolates in grower and finisher diets with RSC was 2.37 and 2.03 μmol/g, respectively, the values being only marginally higher than the maximum recommended dose of glucosinolates (2 μmol/g) for pigs. It is well known that the degradation products of glucosinolates can have negative effects on growth performance, mainly because of a disadvantageous effect on thyroid and liver functions [35]. The compound responsible for antinutritive effect is progoitrin, which forms the toxic compound goitrin. In contrast, a significant improvement in FCR was observed when CSC was added at the level of 3.7% and 7.4% to soybean-based diet for weaned pigs [41]. The enhancement in FCR was ascribed to an increase in the available lysine content and metabolizable energy from CSC. They also noticed that pigs had an initial aversion to the CSC-enriched diet (7.4%) in comparison to the control diet, with a tendency toward a reduction in average daily feed intake (FI). Taranu et al. [50] found no changes in ADG, FI, or FCR when fattening and finishing pigs were fed CSC in comparison to control diet with sunflower meal. The lack of data on the effects on pig growth performance distinctly indicate the need for further research to reach clear understanding about the influence of dietary use of oilseed cakes differing in chemical composition and inclusion level.

### 6.3. Effects on Other Parameters

Kaczmarek et al. [39] observed that dietary inclusion of RSC caused a decrease in carcass lean content and changes in FA profile of backfat, including a rise in the content of MUFA and n-3 PUFA (especially ALA), and a more desirable n-6/n-3 ratio (7.2 vs. 11.5) compared to the control soybean-based diet. In the same study, backfat thickness, meat quality traits, and thyroid gland weight were not affected by the dietary supplementation of RSC. Taranu et al. [50] reported an improvement in biochemical profile with dietary CSC supplementation. Namely, CSC affected the decrease in plasma glucose concentration by 18%, with a tendency to a decrease in plasma cholesterol concentration, and improved plasma antioxidant capacity. CSC also modulated cellular immune response by suppressing spleen proinflammatory markers and increased detoxifying enzyme gene expression. Antioxidant potential of CSC was associated to n-3 PUFA, which modulate secretion of detoxifying enzymes and expression of important markers of inflammation [50]. This was also confirmed by others who found that feeding pigs with CSC may have anticarcinogenic benefits by stimulating the hepatic expression of phase 1 and 2 xenobiotic detoxifying enzymes [41].

## 7. Cold-Pressed Oilseed Cakes in Poultry Nutrition

### 7.1. Effects on Growth Performance

The main effects of the use of cold-pressed cakes in poultry diet are summarized and shown in Appendix A. It was documented that the inclusion of HSC up to 20% in laying hens’ diet did not have an adverse effect on the bird performance [26,30]. More precisely, egg production, FI, FCR, body weight change, laying rate, egg weight, or egg mass were not affected by dietary treatment. In a study by Halle and Schöne [19], laying hens were fed HSC, LSC, or RSC at three inclusion levels (5, 10, and 15%) for six laying months. It was found that FI, FCR, and egg mass production at 15% cake level were significantly lower in comparison to 5 and 10% cake supplementation level. In the same study, the cake inclusion level had a different effect for the three examined cakes. Namely, the FI decreased with increasing RSC level, while no effect of supplementation level was determined in HSC and LSC groups. The egg mass production was lower in LSC compared with HSC groups, while FCR was higher in LSC than other cakes at the 15% inclusion level. When laying hens were fed with a diet containing organically produced SFC (26% dietary level), it was revealed that hen performance was not significantly affected by dietary treatment [27]. SFC contains considerable amounts of crude fiber and insoluble non-starch polysaccharides, which are believed to exert beneficial effects on chicken performance and small intestinal health by stimulating gizzard activity and activity of digestive enzymes [28]. Some studies revealed that formulating diets for laying hens and broilers with up to 10% CSC did not impair production performance parameters [51,85,86,87]. The inclusion level of camelina seed and its products in poultry diet is usually not higher than 15%, because products from glucosinolate metabolism can deteriorate diet palatability and animal performance [37]. In a study by Ryhänen et al. [33], 1-day-old broilers were fed starter diet supplemented with 5 and 10% CSC for 37 days. It was observed that CSC depressed FI during early growth (1–14 days) and impaired FCR during the starter period and throughout the entire experiment period (1–37 days). Furthermore, an increasing inclusion level caused linear reduction in the growth of the birds between 15 and 37 days of age, whereas the average final BW of birds was reported to be 7–10% lower than that of birds receiving control diet. Authors suggested that the reason for the impaired performance could be the lower ME of CSC than those in SBM-based control diet and a high concentration of glucosinolates and their breakdown products in CSC. On the other hand, in the same research, no significant enlargement of the thyroid gland or liver lesions, which are usually related to high dietary glucosinolates concentration, was observed. Pekel et al. [88] conducted an experiment with broiler chicken which were fed with increasing levels (12, 24, and 36%) of two different CSC (CSC 1 and CSC 2). The results showed that growth BWG and FCR were negatively affected with increasing dietary level of both CSC, while a more severe effect on growth performance was observed when the birds were receiving graded levels of CSC 1. The differences in the effect of CSC might arise from the use of different camelina varieties grown under different climatic zones and fertilization programs, which influenced variations in their chemical composition (especially antinutritive factors) [85]. The reduced BWG in CSC 1 was partially attributed to the decrease in FI, as the reduction in FI of nearly 30% was observed when the dietary inclusion level of CSC 1 increased from 12 to 36%. The concentration of glucosinolate fraction (9-methylsulfiniylnonyl) was significantly higher in CSC 1 (7.55 µmol/g) than in CSC 2 (5.97 µmol/g), which was attributed to severe deterioration in FI with the inclusion of CSC 1 in the diet.

### 7.2. Effects on Blood Parameters

The effect of feeding chickens with cold-pressed cakes on blood biochemistry was only found in a few studies dealing with dietary inclusion of CSC. Feeding broiler chickens with 8% CSC beneficially altered blood biochemistry profile by decreasing the plasma concentration of glucose by 4.4% and reducing total cholesterol (12.3%) and its fractions of high-density lipoprotein cholesterol and low-density lipoprotein cholesterol by 9.2 and 29.5%, respectively [51]. The decrease in plasma total cholesterol was related to hypolipidemic and hypocholesterolemic properties of PUFA in CSC due to a suppression of hepatic cholesterol production in the liver. Dietary PUFA regulate transcriptional activity of specific nuclear receptors, thus controlling transcription of specific genes that take part in hepatic carbohydrate and lipid metabolism [51,89]. The concentration of very low-density lipoprotein cholesterol and triglycerides were not significantly affected by dietary treatment [51]. In a similar study, feeding broiler chickens with feed mixture supplemented with 10% CSC did not have a significant effect on serum lipid status, regardless of the tendency towards a decrease in the plasma content of cholesterol, its fractions, and triglycerides [89]. CSC contains glucosinolates, which can exert a deleterious effect on the functions of the thyroid gland and induce hypertrophy of this gland and decrease in the content of plasma thyroid hormones tyroxine (T4) and triiodothyronine (T3). However, the content of T3 and T4 in blood showed to be unaffected by dietary inclusion of CSC [89].

### 7.3. Effects on Quality and FA Profile of Egg and Meat

Regarding the quality of animal products, some studies reported that the supplementation of laying hens’ diet with HSC (up to 20%) did not have negative effects on egg quality [26,30]. It was suggested that HSC did not contain substances that may impair egg quality and that nutritive compounds in the HSC were available to the chickens [26]. In contrast, an increasing dietary level of HSC, RSC, and LSC (from 5 to 15%) decreased the yolk percentage and increased the egg white percentage [19]. In the same study, egg shell percentage was higher for hens fed RSC than those fed LSC, and egg shell content was higher for 15% dietary inclusion compared to that for 5% cake level. Orczewska-Dudek et al. [87] observed that yolks from hens fed a CSC-supplemented diet had higher color index, whereas egg shell weight, egg shell thickness, density, and proportion of egg shell in the egg were unaffected by the dietary treatment. It was also noted that CSC did not have any adverse effect on the sensory profile of eggs [87]. More precisely, palatability and flavor of eggs from hens fed CSC did not differ from those of the control group containing rapeseed oil. Furthermore, Ryhänen et al. [33] demonstrated that the inclusion of CSC in broilers’ diet did not show a deleterious effect on the sensory quality (taste, juiciness, and tenderness) of broiler meat. However, others reported that the supplementation of the broiler chicken diet with 10% CSC had less favorable influence on the sensory properties by deteriorating the tastiness and flavor of the cooked meat compared with the control group, whereas the addition of camelina seed oil improved the juiciness of meat [85].

The inclusion of cakes in the feed mixture is a way to modulate the FA profile of animal-derived products and achieve desirable PUFA for human nutrition. Dietary inclusion of cold-pressed cake can produce eggs and meat with lower SFA and MUFA but higher PUFA content [19,30,85,87]. The supplementation of laying hens’ diet with increasing level of RSC, LSC, and HSC caused a linear increase in PUFA and decrease in SFA and MUFA in yolk lipids [19]. The rise in PUFA was the consequence of the increase in LA and ALA, regardless of the protein source. The highest share of LA was observed when hens were fed 15% HSC (25.5%), while the highest level of ALA was obtained when hens were offered 15% LSC (5.2%). Furthermore, the ratio of n-6/n-3 decreased from 16.8 to 11.1 for RSC, from 8.5 to 4.4 for LSC, and from 11.8 to 8.4 for HSC [19]. In a study by Mierliță [30], dietary inclusion of HSC resulted in a lower share of MUFA but higher share of PUFA than the control diet, similar to that observed in the diet supplemented with hemp seed. Furthermore, Mierliță [30] also noticed an increase in n-6 and n-3 PUFA but reduction in n-6/n-3 ratio from 11.1 to 4.2. Similar but less pronounced observations were reported in other studies dealing with dietary inclusion of CSC [87]. The authors observed lower n-6/n-3 ratio (3.2) in egg yolk with CSC supplementation than that from rapeseed-oil-based diet (5.3) but similar when camelina seed oil was added to a hens’ diet (2.9). Additionally, Ryhänen et al. [33] documented a significant decrease in SFA and MUFA and increase in PUFA (particularly LA and ALA) for both female and male broilers when the dietary content of CSC was raised from 0 to 10%. They also reported a decrease in n-6/n-3 ratio compared to a control diet from 4.6 to 2.7 for female broilers and from 5.1 to 2.7 for male broilers. The results agree with the results of others who observed a decline in n-6/n-3 ratio from 4.1 to 2.5 with dietary supplementation of CSC (10%), as well as increase in PUFA (LA and ALA) and decrease in MUFA [85]. The summarized results showed that a favorable n-6/n-3 ratio in egg yolk and in broiler meat was achieved in the studies which used HSC, LSC, and CSC [30,33,85,87], which was expected, since these cakes are especially rich in essential n-3 and n-6 PUFA. CSC contains erucic acid, which is known to inhibit growth and cause detrimental changes in animals’ organs, thus representing a health risk [33]. However, when feeding chickens with CSC-supplemented diet, the content or erucic acid in egg yolk or breast meat was very low (around 0.1%), which indicated that these products were safe for human consumption [33,85]. It has been considered that meat of animals containing high content of PUFA is prone to oxidative processes, which has a detrimental effect on its sensory properties and shelf life [85]. In the study of Untea et al. [86], the supplementation of 3% CSC in broilers’ diet improved oxidative stability of breast muscle. CSC also contains high levels of natural antioxidants, such as tocopherols, tocotrienols, phenolic compounds, and phytosterols, thus limiting PUFA oxidation in the breast muscle [85]. In contrast, other studies reported no significant changes in oxidative stability of breast muscle of broilers fed a diet containing CSC [85]. In contrast to HSC, LSC, RSC, and CSC, available information on the influence of SFC and PSC on the quality and FA profile of meat and eggs are lacking in the scientific literature.

## 8. Conclusions

The exploitation of cold-pressed oilseed cakes, which generate in a large quantity as the by-products of the oil industry, is recognized as a feasible and crucial point in promoting environmental sustainability and eco-friendly concept on one hand, as well as providing a cheap and valuable feed ingredient that could diminish the dependence on non-sustainable soybean meal on the other hand. Cold-pressed cakes from rapeseed, hempseed, linseed, sunflower seed, camelina seed, and pumpkin seed have shown to be promising alternatives to conventionally used protein feed ingredients in animal feeding. The cakes are distinguished by high nutritive values, as they contain appreciable amount of crude protein and essential amino acids and high lipid content, which make them a suitable protein and energy source for ruminants and nonruminants. Furthermore, cakes as a feed ingredient in animal diet may be efficiently utilized as a source of essential fatty acids, particularly for the production of animal products with n-3 and n-6 PUFA, which have demonstrated numerous beneficial health effects. However, the summarized effects on growth performance were contradictory and depended on the type and nutritional profile of cake, supplementation level, animal category, experiment duration, etc. Special attention should be focused on antinutritional compounds that can limit feed intake and nutrient utilization and, hence, restrict the inclusion of cakes in feed formulations for pigs and poultry. Therefore, in order to acquire depth knowledge on the effects and establish the acceptable dietary supplementation level of cold-pressed cakes without any adverse influence on animal, further investigations including various categories of animals at different growing stages are required.

## Figures and Tables

**Figure 1 foods-12-00432-f001:**
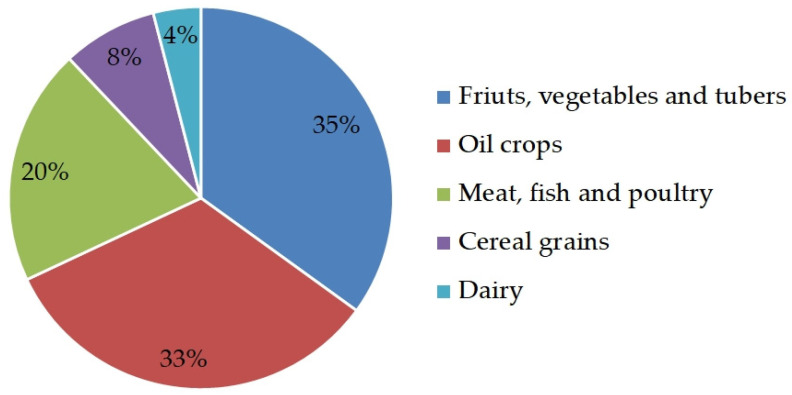
Estimation of food waste produced per sub-sectors at the processing stage in Europe (adapted from Rao, Bastand de Boer [4]).

**Figure 2 foods-12-00432-f002:**
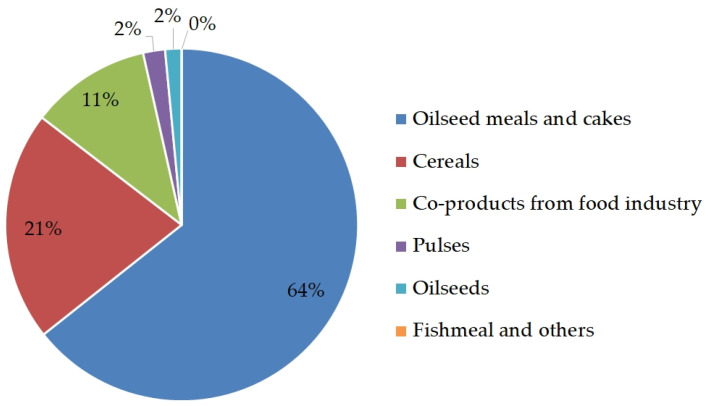
The use of protein materials by the animal feed sector in Europe (adapted from FEDIOL [14]).

**Table 1 foods-12-00432-t001:** Chemical composition (on DM basis) of cold-pressed oilseed cakes.

Nutrient	RSC	HSC	LSC	SFC	CSC	PSC
Protein (%)	19.4 [15]–45.0 [16]	24.8 [17]–36.1 [18]	32.2 [19]–35.9 [20]	21.6 [5]–37.7 [21]	31.3 [22]–42.0 [23]	38.3 [17]–62.3 [24]
Fat (%)	9.6 [16]–31.3 [10]	8.9 [25]–16.4 [26]	11.5 [21]–21.4 [17]	11.7 [27]–31.4 [21]	10.5 [23]–26.0 [22]	9.0 [24]–36.2 [17]
Ash (%)	4.2 [15]–8.1 [16]	6.3 [25]–7.5 [17]	4.9 [18]–5.9 [20]	4.3 [28]–6.3 [8]	4.5 [29]–6.5 [23]	7.5 [15]–8.1 [21]
Fiber (%)	6.5 [16]–19.5 [15]	25.1 [30]–26.1 [25]	8.6 [31]–9.5 [32]	12.6 [5]–37.0 [28]	11.3 [33]–17.0 [34]	23.1 [15]
NDF (%)	17.0 [35]–33.7 [36]	37.7 [18]–53.4 [17]	17.6 [18]–21.3 [20]	n.d	23.7 [37]–43.4 [29]	11.8 [17]–13.9 [24]
ADF (%)	9.0 [35]–21.0 [36]	29.7 [30]–39.2 [17]	13.0 [19]–14.2 [17]	n.d.	11.1 [38]–21.6 [29]	5.1 [17]–11.1 [24]
Glucosinolates, (µmol/g)	8.8 [35]–16.9 [39]	n.d.	n.d.	n.d.	22.9 [33]–46.1 [37]	n.d.

RSC—rapeseed cake; HSC—hempseed cake; LSC—linseed cake; SFC—sunflower seed cake; CSC—camelina seed cake; PSC—pumpkin seed cake; NDF—neutral detergent fiber; ADF—acid detergent fiber; n.d.—non defined. Figures between square brackets [...] indicate references.

**Table 2 foods-12-00432-t002:** Amino acid (AA) profile (% as-fed basis) of cold-pressed oilseed cakes.

Essential AA (%)	RSC	HSC	LSC	SFC	CSC	PSC
Arginine	1.4 [10]–1.9 [18]	4.1 [18]	3.0 [18]	n.d.	2.5 [37]–3.5 [23]	n.d.
Histidine	0.6 [10]–0.8 [18]	1.0 [18]	0.8 [18]	n.d.	0.8 [37]–1.1 [47]	n.d.
Isoleucine	1.0 [10]–2.6 [16]	1.4 [18]	1.5 [18]	n.d.	1.1 [29]–1.6 [23]	n.d.
Leucine	1.7 [10]–2.0 [18]	2.3 [18]	2.0 [18]	n.d.	2.1 [37]–2.7 [23]	n.d.
Lysine	1.4 [10]–1.6 [18]	1.3 [18]	1.3 [18]	n.d.	1.4 [37]–2.1 [23]	n.d.
Methionine	0.5 [10]–0.6 [16]	0.8 [18]	0.6 [18]	n.d.	0.3 [29]–0.7 [23]	n.d.
Phenilalanine	1.0 [10]–1.5 [16]	1.6 [18]	1.5 [18]	n.d.	1.3 [29]–1.7 [23]	n.d.
Threonine	1.0 [10]–1.3 [18]	1.9 [18]	1.2 [18]	n.d.	1.3 [37]–1.6 [23]	n.d.
**Nonessential AA (%)**						n.d.
Valine	1.3 [10]–2.3 [16]	1.8 [18]	1.7 [18]	n.d.	1.6 [37]–2.2 [23]	n.d.
Alanine	1.1 [10]–1.5 [16]	1.5 [18]	1.5 [18]	n.d.	1.4 [33]–1.9 [23]	n.d.
Aspartic acid	1.7 [10]–2.8 [16]	3.5 [18]	3.1 [18]	n.d.	2.5 [33]–3.4 [23]	n.d.
Cysteine	0.6 [10]–1.2 [16]	0.6 [18]	0.7 [18]	n.d.	0.3 [47]–0.9 [23]	n.d.
Glutamic acid	3.7 [10]–7.8 [16]	5.8 [18]	6.2 [18]	n.d.	5.3 [37]–6.8 [23]	n.d.
Glycine	1.2 [10]–2.1 [16]	1.5 [18]	1.9 [18]	n.d.	1.6 [37]–2.1 [23]	n.d.
Proline	1.5 [10]–1.6 [18]	1.4 [18]	1.2 [18]	n.d.	1.5 [29]–2.1 [47]	n.d.
Serine	0.8 [16]–1.3 [18]	1.7 [18]	1.5 [18]	n.d.	1.3 [47]–1.8 [29]	n.d.
Tyrosine	0.7 [10]–1.0 [18]	1.3 [18]	0.9 [18]	n.d.	0.9 [29]–1.1 [23]	n.d.

RSC—rapeseed cake; HSC—hempseed cake; LSC—linseed cake; SFC—sunflower seed cake; CSC—camelina seed cake; PSC—pumpkin seed cake. n.d.—non defined.

**Table 3 foods-12-00432-t003:** Fatty acid (FA) profile of cold-pressed cakes (% of total fatty acids).

Fatty Acids	RSC	HSC	LSC	SFC	CSC	PSC
Palmitic acid	4.4 [19]–4.8 [48]	6.0 [19]–9.3 [49]	5.9 [19]	5.5 [5]	7.1 [50]–7.4 [51]	12.3 [15]–13.2 [24]
Stearic acid	0.6 [19]–1.6 [48]	1.6 [19]–3.8 [49]	2.9 [19]	1.1 [5]	2.0 [51]	4.9 [24]–5.2 [15]
Oleic acid	51.6 [48]–59.6 [19]	11.4 [19]–13.1 [49]	22.1 [19]	10.3 [5]	15.6 [50]–18.8 [52]	28.8 [15]–29.6 [24]
Linoleic acid (LA)	20.2 [36]–23.5 [19]	52.5 [49]–56.0 [25]	17.1 [19]	32.8 [5]	19.4 [53]–25.0 [50]	49.0 [24]–51.2 [15]
α-linolenic acid (ALA)	9.4 [36]–10.6 [19]	14.4 [25]–24.7 [19]	51.5 [19]	0.2 [5]	22.0 [52]–35.0 [53]	0.5 [15]–0.6 [24]
PUFA	29.6 [36]–34.1 [19]	70.4 [25]–80.5 [19]	68.6 [19]	34.2 [5]	44.7 [52]–60.5 [50]	47.0 [24]–52.9 [15]
n-3	9.4 [36]–10.6 [19]	14.4 [25]–24.7 [19]	51.5 [19]	0.2 [5]	22.0 [52]–32.9 [50]	1.3 [15]–1.7 [24]
n-6	20.2 [36]–23.5 [19]	53.5 [30]–56.0 [25]	17.1 [19]	34.0 [5]	21.1 [51]–27.7 [50]	45.1 [24]–51.7 [15]
n-6/n-3	2.2 [36]–2.4 [48]	2.3 [19]–3.9 [25]	0.3 [19]	170 [5]	0.7 [51]–1.0 [52]	26.5 [24]–39.2 [15]

RSC—rapeseed cake; HSC—hempseed cake; LSC—linseed cake; SFC—sunflower seed cake; CSC—camelina seed cake; PSC—pumpkin seed cake; PUFA—polyunsaturated fatty acids.

## Data Availability

No new data were created or analyzed in this study. Data sharing is not applicable to this article.

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
