# Peer review of "Cold-Pressed Oilseed Cakes as Alternative and Sustainable Feed Ingredients: A Review"

_foods, 2023, doi:10.3390/foods12030432_

Round 1

Reviewer 1 Report

Dear authors, the current paper, entitled Cold-pressed oilseed cakes as alternative and sustainable feed ingredients: A review is very well structured, with satisfactory results, with a lot of data review and is a pleasure to be read.

However, I have some observations, which I consider can help to improve the presentation and some suggestions were the authors can add the missing information which I think they missed.  

Figure 1 and Figure 2 are missing from the paper. Also, both figures are placed wrong in the manuscripts. Figure 1 must be placed under subchapter 2 and figure 2 under subchapter 3.

Rows 70 to 77. Indeed, the authors are right about the two used terms, cakes and meals, however, in some countries, they use old techniques for oil extractions, and the by-product resulted is named meal. Some of the oil extraction producers are able to make a cake or a meal from the wastes resulted, and the process is called mechanical pressing for oil extraction.  However, I just want to point out, that although there is confusion among researchers about the difference between the two by-products used, not all of them are wrong.  

In table 3, why the authors don’t report the total PUFA and the total of the n-3 and n-6, as well as the ratio of n-6/n-3, since these parameters are very important for both animals and humans, and have great implications as the authors reported in the paper.

Row 60. Provide the year for FAO, and place it as a reference.

Row 66, the studied by-products are not mainly used as sources of protein. For example linseed/flaxseed, cottonseed, rapeseed meal are mostly used as dietary ingredients in poultry and pigs to enhance the product quality, as the large data shows (https://doi.org/10.3390/nu14091969, https://doi.org/10.1038/s41598-021-00343-1 , https://doi.org/10.1080/87559129.2019.1584817 https://doi.org/10.1016/j.smallrumres.2016.01.001

Row 373,377, 387, 400, 404, 408, 414, 415, 423 the refence is cited wrong, please check

Row 542, the refence is cited wrong, please check

Row 562, same observation.

Row 565 and 572, same observation

Row 602-603. Indeed, higher levels of the reviewed by-products used in poultry can affect the production performances, but beside that, in a recent study was reported that flaxseed and rapeseed up to 9% can produce undesirable effects on egg quality. The experimental duration of the feeding also have a major influence corroborated with the inclusion level (https://doi.org/10.1038/s41598-021-00343-1 )

Row 606, same observation, please check all cited sources and write them accordingly.

Row 583. Section 7 is entitled Cold-pressed oilseed cakes in poultry nutrition and the subsection is Effects on growth performance, the authors mainly explained the effects on laying hens, not on broilers, so correctly it will be to change the subheading name in Effect on production performances, and to add some effect on broilers also.

Some suggestions are below and in the literature many studies can be found

https://doi.org/10.1080/09712119.2014.978773

https://doi.org/10.3382/ps.2009-00051

https://doi.org/10.3382/ps/pew371

https://doi.org/10.1111/jpn.12654

In section 7.3. Effects on quality and FA profile of egg and meat, I have noticed that the pumpkin is not discussed at all. Also, in Supplementary table 3, there is no word about the effect of pumpkin meal/cake on poultry. Some suggestions can be found below also. However, the authors must consider the type of pumpkin used. For example these authors used Cucurbita moschata in laying hens diets (doi: 10.5713/ab.21.0044 ) others used Cucurbita pepo (DOI: 10.9734/AJAAR/2017/35742). Although there are not so many studies regarding the use of PCM on poultry as the authors found, but some references are available to be reviewed.

Rows 665, 663, 687 the references are not correctly cited, please check.

General comment. All the cited references should be checked and placed accordingly in the text. Update the missing information, especially for PSC, where is little explained and discussed.

Author Response

We sincerely thank the reviewer for the effort and time put into the review of this manuscript and for constructive remarks. All suggestions and comments are carefully considered point by point and responded accordingly.

Reviewer: Figure 1 and Figure 2 are missing from the paper. Also, both figures are placed wrong in the manuscripts. Figure 1 must be placed under subchapter 2 and figure 2 under subchapter 3.

Authors: In both submitted versions of the manuscript (Word and PDF) Figures 1 and 2 are visible and placed properly in the text. Figure 1 is placed under subchapter 1 because data presented in Figure 1 are described within this subchapter, while Figure 2 describes the use of protein materials by the animal feed sector and is explained within subchapter 2.

Reviewer: Rows 70 to 77. Indeed, the authors are right about the two used terms, cakes and meals, however, in some countries, they use old techniques for oil extractions, and the by-product resulted is named meal. Some of the oil extraction producers are able to make a cake or a meal from the wastes resulted, and the process is called mechanical pressing for oil extraction.  However, I just want to point out, that although there is confusion among researchers about the difference between the two by-products used, not all of them are wrong. 

Authors: With this part of manuscript we wanted to address the problem in terminology that evidently exists in science literature. There is a clear difference between cake and meal not only based on obtaining method, but also based on their chemical composition. We wanted to emphasize that and propose some kind of definitions so other authors can use proper terms when writing papers about their future research. In this way, the comparison of the obtained results and material used in the research (cake or meal) will be more feasible and any confusion about used terms will be avoided.

Reviewer: In table 3, why the authors don’t report the total PUFA and the total of the n-3 and n-6, as well as the ratio of n-6/n-3, since these parameters are very important for both animals and humans, and have great implications as the authors reported in the paper.

Authors: Reviewer suggestion to complement Table 3 with data on total PUFA, n-6, n-3, n-6/n-3 has been accepted and hence Table 3 has been modified.

Reviewer: Row 60. Provide the year for FAO, and place it as a reference.

Authors: Due to a technical issue with Endnote (reference manager) we were not able to add new reference at this point. Therefore FAO reference was deleted.

Reviewer: Row 66, the studied by-products are not mainly used as sources of protein. For example linseed/flaxseed, cottonseed, rapeseed meal are mostly used as dietary ingredients in poultry and pigs to enhance the product quality, as the large data shows (https://doi.org/10.3390/nu14091969, https://doi.org/10.1038/s41598-021-00343-1, https://doi.org/10.1080/87559129.2019.1584817, https://doi.org/10.1016/j.smallrumres.2016.01.001

Authors: We didn’t change the mentioned sentence because it is not completely incorrect. Studied by-products are being used as a source of protein in animal nutrition, in addition to being rich in oil, essential fatty acids, amino acids etc. Beneficial effects of using cold pressed cakes in animal nutrition (especially on FA composition of milk, meat and eggs) are thoroughly elaborated through the whole manuscript.   

Reviewer: Row 373,377, 387, 400, 404, 408, 414, 415, 423 the reference is cited wrong, please check

Authors: The references are doubled checked and we didn’t find any mistake.

Reviewer: Row 542, the reference is cited wrong, please check

Authors: The reference is thoroughly checked and authors found it was properly cited.

Reviewer: Row 562, same observation.

Authors: Authors thoroughly checked the reference and found it was properly cited.

Reviewer: Row 565 and 572, same observation

Authors: Authors thoroughly checked the references and corrected the reference numbered 38.

Reviewer: Row 602-603. Indeed, higher levels of the reviewed by-products used in poultry can affect the production performances, but beside that, in a recent study was reported that flaxseed and rapeseed up to 9% can produce undesirable effects on egg quality. The experimental duration of the feeding also have a major influence corroborated with the inclusion level (https://doi.org/10.1038/s41598-021-00343-1 )

Authors: Thank you for this suggestion. However, we didn’t take this article in consideration because flaxseed meal was used in combination with sea buckthorn meal, and rapeseed meal in combination with grapeseed meal. Therefore, the influence on egg quality could not be attributed to a particular meal but to a combination of two used meals.

Reviewer: Row 606, same observation, please check all cited sources and write them accordingly.

Authors: Authors thoroughly checked the reference and found it was properly cited

Reviewer: Row 583. Section 7 is entitled Cold-pressed oilseed cakes in poultry nutrition and the subsection is Effects on growth performance, the authors mainly explained the effects on laying hens, not on broilers, so correctly it will be to change the subheading name in Effect on production performances, and to add some effect on broilers also.

Some suggestions are below and in the literature many studies can be found

https://doi.org/10.1080/09712119.2014.978773

https://doi.org/10.3382/ps.2009-00051

https://doi.org/10.3382/ps/pew371

https://doi.org/10.1111/jpn.12654

Authors: Section 7.1 encompasses the effects on both laying hens and broilers. References numbered 27, 32, 87, 50, 84 and 85 refer to the effects of cold-pressed cakes on broiler growth performance and were included in this section. We appreciate an effort by the reviewer to upgrade the cited literature, however, we observed that suggested literature don’t refer to the use of cold-pressed cakes in animal nutrition as is the aim of the presented review.

Reviewer: In section 7.3. Effects on quality and FA profile of egg and meat, I have noticed that the pumpkin is not discussed at all. Also, in Supplementary table 3, there is no word about the effect of pumpkin meal/cake on poultry. Some suggestions can be found below also. However, the authors must consider the type of pumpkin used. For example these authors used Cucurbita moschata in laying hens diets (doi: 10.5713/ab.21.0044) others used Cucurbita pepo (DOI: 10.9734/AJAAR/2017/35742). Although there are not so many studies regarding the use of PCM on poultry as the authors found, but some references are available to be reviewed.

Authors: By a detailed literature search, the authors didn’t find any information about the influence of dietary inclusion of cold-pressed pumpkin seed cake on quality, FA profile of eggs or meat. As outlined in chapter 3, the present manuscript was focused only on the use of cakes that were obtained by cold-pressing method. Therefore, other papers related to the use of pumpkin seed meal or papers with no information about the pressing conditions applied were excluded from the scoping review (i.e. doi: 10.5713/ab.21.0044, doi:10.9734/AJAAR/2017/35742). However, in order to point out the absence of information about the effects of PSC on quality and FA profile of egg and meat, the authors provided the information in the manuscript.

Reviewer: Rows 665, 663, 687 the references are not correctly cited, please check.

Authors: Authors thoroughly checked the references and found they were properly cited.

Reviewer: General comment. All the cited references should be checked and placed accordingly in the text. Update the missing information, especially for PSC, where is little explained and discussed.

Authors: All the references are cautiously checked throughout the entire manuscript and placed accordingly in the text. The lack of presented information about the use of PSC is explained in the previous comment.

Reviewer 2 Report

Regarding review entitled " Cold-pressed oilseed cakes as alternative and sustainable feed 2 ingredients: A review’’ This in an interesting review, however I have some comments:

L21. Please rephrase

L27. ‘’the acceptable dietary level without delivering negative effects on animal’’ I think there are already published papers that have been studied the graded levels of these oilseed cakes or meals

Why Table S1 is supplementary, it should be presented in the review, and for other supplementary tables. The authors should present these tables in the review. 

L257. This subheading only mentions growth performance, while the contents also include milk yield, composition, and digestibility, please revise

Each section should have two or three sentences to conclude what the authors discussed in the section to be clear for the readers.

L633. Blood is enough, delete the plasma

Author Response

We sincerely thank the reviewer for the effort and time put into the review of this manuscript and for constructive remarks. All suggestions and comments are carefully considered point by point and responded accordingly.

Reviewer: L21. Please rephrase

Author: The sentence has been rephrased.

Reviewer: L27. ‘’the acceptable dietary level without delivering negative effects on animal’’ I think there are already published papers that have been studied the graded levels of these oilseed cakes or meals

Authors: The sentence has been deleted.

Reviewer: Why Table S1 is supplementary, it should be presented in the review, and for other supplementary tables. The authors should present these tables in the review.

Authors: We had some technical problems to insert those 3 tables in main text. We also think that all supplementary tables should be inserted in main text, so we will ask journal technical support to help us with that.

Reviewer: L257. This subheading only mentions growth performance, while the contents also include milk yield, composition, and digestibility, please revise

Authors: The title of the subheading has been changed as suggested.

Reviewer: Each section should have two or three sentences to conclude what the authors discussed in the section to be clear for the readers.

Authors: We put a great effort to summarize the results and draw a straightforward conclusion for each animal category and oilseed cake type. However, conflicting results were observed depending on the cake type and chemical composition (protein, lipid content, presence of antinutritional factors, amino acid profile, fatty acid composition, etc.), dietary inclusion level, animal category, and experiment duration. Despite the limited number of papers dealing with particular effects and that many factors were observed, we did our best to highlight the most relevant results and findings for each section.

Reviewer: L633. Blood is enough, delete the plasma

Authors: It has been corrected as suggested.

Round 2

Reviewer 1 Report

Dear Authors,

I agree with your comments.

However, the refecrences are still wrong cited. For example

row 245 Moloney, Woods and Crowley [21] should be Moloney et al [21]

row 255-256 Benhissi, García Rodríguez and Beltrán de Heredia [53], should be Benhissi et al., 53]

Benhissi, Beltrán de Heredia and García Rodríguez [54] should Benhissi et al 54]

row 307 same - Lerch, Ferlay, Pomiès, Martin, Pires and Chilliard [60].

row 338 Amores, Virto, Nájera, Mandaluniz, Arranz, Bustamante, Valdivielso, Ruiz de Gordoa, García-Rodríguez, Barron and de Renobales [65].

row 376 Šalavardić, Novoselec, Đidara, Steiner, Ćavar, Modić Šabić and Antunović [24]

row 380 Goiri, Zubiria, Lavín, Benhissi, Atxaerandio, Ruiz, Mandaluniz and García-Rodríguez [12]

row 385 Mihhejev, Henno, Ots, Rihma, Elias, Kuusik and Kärt [58]

row 392 Jozwik, Strzalkowska, Bagnicka, Lagodzinski, Pyzel, Chylinski, Czajkowska, Grzybek, Sloniewska, Krzyzewski and Horbańczuk [68]

row 395 Boldea, Dragomir, Gras and Ropota [14]

row 398 studies Mihhejev, Henno, Ots, Rihma, Elias, Kuusik and Kärt [58]. (in thi case and other is no need to give the authors name, the indication is enough [58].

and so on. This are those mistakes that I have signaled previously. 

Best of luck!